computational chemistry/biochemistry/ spectroscopy

oleanolic, ursolic, DFT, vibrational spectroscopy

**Authors for correspondence:**
Rafael C. González-Cano
e-mail: rafacano@uma.es
Antonio Heredia
e-mail: heredia@uma.es

This article has been edited by the Royal Society of Chemistry, including the commissioning, peer review process and editorial aspects up to the point of acceptance.

# Structure determination of oleanolic and ursolic acids: a combined density functional theory/vibrational spectroscopy methodology

Luz D. M. Gómez-Pulido[1], Rafael C. González-Cano[2], Eva Domínguez[1] and Antonio Heredia[3]

[1]IHSM La Mayora, Departamento de Mejora Genética y Biotecnología, Consejo Superior de Investigaciones Científicas, E-29750 Algarrobo-Costa, Málaga, Spain
[2]Departamento de Química Física, Facultad de Ciencias, and [3]IHSM La Mayora, Departamento de Biología Molecular y Bioquímica, Universidad de Málaga, E-29071 Málaga, Spain

RCG-C, 0000-0002-2165-6469

Raw samples of oleanolic and ursolic acids, a class of terpenoid acids mainly found in the leaf and fruit cuticles of some plant species, can be defined as a blend of clusters of different conformers aggregated in dimers and tetramers by means of hydrogen bonds and stabilized by non-electrostatic interactions.

## 1. Introduction

The outer surface of epidermal plant cell walls is covered by an extracellular and continuous membrane called the cuticle [1]. Cutin, an insoluble amorphous polymer matrix of interesterified polyhydroxy fatty acids, is the main component of the cuticle. Cuticular waxes, which can be embedded within the cuticle (intracuticular) or deposited on the outer surface (epicuticular), are the other lipid component of the plant cuticle [2]. Waxes are a complex mixture of very long-chain alkanes, alcohols, fatty acids and triterpenoids acids, usually present in variable proportions [3]. The epicuticular wax layer [4] is described, from the molecular point of view, as a mixture of both crystalline and amorphous regions [3,5]. The crystallinity of the outer part of the cuticle is related to the physical and biological behaviour of the cuticle and, hence, with some of their main properties and functions [6].

One of the main roles of waxes is to regulate water and gas exchange with the environment, acting, together with the cutin matrix, as a physical barrier limiting the movement of water

and other molecules across the plant–atmosphere interface [7]. Additionally, they attenuate UV radiation and provide mechanical support and resistance against pests [8].

Oleanolic and ursolic acids are pentacyclic triterpenoids present in many leaf and fruit cuticles [9–13]. In fact, the cuticle is the main natural source of these compounds, where they have been associated with the semicrystalline region of plant waxes [14]. Chemical analysis of these terpenoid acids has shown different crystalline and semicrystalline forms depending on the solvent [15] and the thermal treatment [16] employed. This solid state crystallization can be a major concern given their potential application in pharmaceutical formulations [17–19] and medical applications [20–26]. Recent molecular modelling studies have suggested that these terpenoid acids could act as inhibitors against the main protease of SARS-CoV-2 [27].

In order to complete our understanding of the molecular structure of oleanolic and ursolic acids, theoretical calculations have been carried out using the density functional theory (DFT) method. Results and further discussion are accompanied by the corresponding experimental Fourier transform infrared spectroscopy (FTIR) spectra and additional experimental data of these molecules.

# 2. Methodology

## 2.1. Computational details

DFT calculations were performed with Gaussian 16 software [28] using the B3LYP functional together with the 6–31G** basis set. This is a hybrid functional combining the Hartree–Fock and Becke exact exchange functionals [29,30] with the Lee–Yang–Parr correlation functional (LYP) [31]. It has been widely employed in geometric optimizations and in the evaluation of vibration frequencies. An empirical dispersion correction GD3 was used for the analysis of long-range intermolecular interactions [32]. Structures were optimized within an $n$-octanol environment using the Polarizable Continuum Model in order to mimic the average polarity present in the cutin matrix [33]. Theoretical Infrared spectra were constructed after calculation of the vibrational normal modes using a FWHM (Full Width at Half Maximum) of 10 cm$^{-1}$. Calculations were carried out in the Supercomputing and Bioinnovation Center (SCBI) of the University of Málaga.

Graphic editing of the optimized structures was done with the Chimera 1.11.2 software [34] and intermolecular distances were measured with Mercury 3.9 [35,36].

The relative binding energy (RBE) allows us to compare the stabilization of each aggregate with the corresponding monomeric species. This parameter can be calculated as

$$\text{RBE} = \frac{E_a - n \cdot E_m}{n},$$

where $E_a$ is the potential energy for $n$ aggregated molecules and $E_m$ is the potential energy for the isolated monomer.

## 2.2. Fourier transform infrared spectroscopy

FTIR spectra were recorded with a Bruker Tensor27 FT-IR spectrophotometer. Samples were prepared using the KBr pellet procedure without previous preparation. Spectra were collected within the 4000–400 cm$^{-1}$ range with a 4 cm$^{-1}$ resolution and 64 accumulations per sample, using the air as blank. For variable temperature measurements, a Specac Cell model GS21525 coupled with a Graseby Specac automatic temperature control system that allows working in the range −170°C to +250°C was employed. Baseline correction was performed with OPUS 6.5 software.

# 3. Results and discussion

## 3.1. Molecular structure analysis of oleanolic and ursolic monomers

Oleanolic acid (3-β-Hydroxyolean-12-en-28-oic acid) and ursolic acid (3-β-Hydroxyurs-12-en-28-oic acid) structures are based on a carboxylic functionalization on C-10 of β- and α-amyrin, respectively, [37] as it can be observed in figure 1a. The structural analysis of their respective monomers, named OLE$_{\text{mon}}$ and URS$_{\text{mon}}$, was performed within an $n$-octanol environment.

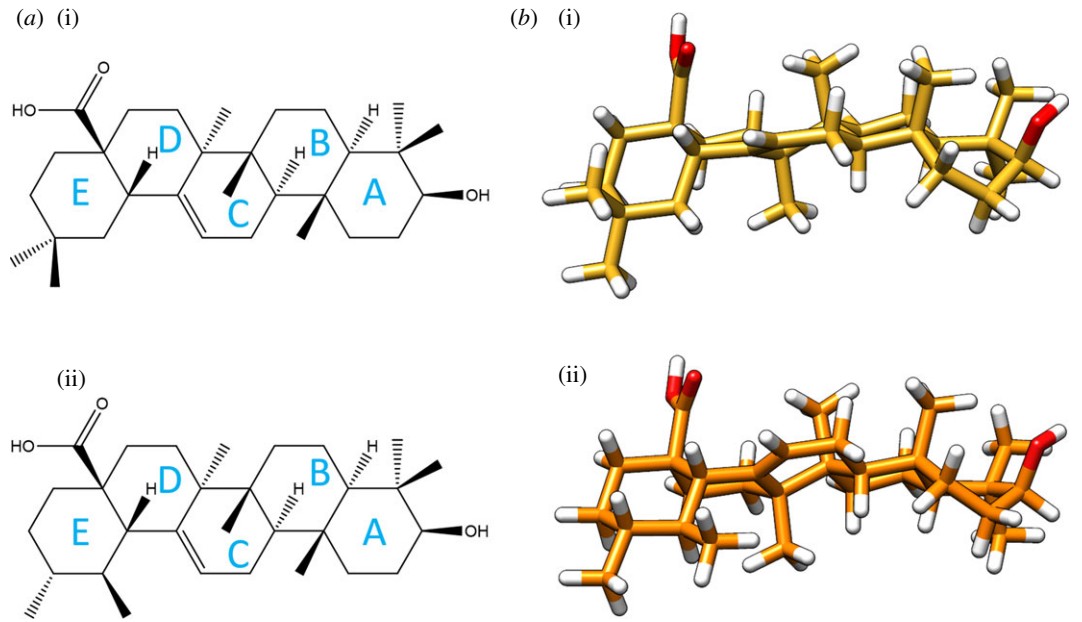

**Figure 1.** Representation of the oleanolic (i) and ursolic (ii) acids (*a*) and their corresponding optimized structures (*b*).

As it has been previously reported for both molecules [2], the hydroxyl group displayed a 62° angle with respect to the main molecular plane, due to the boat conformation adopted by the A ring. Additionally, the E ring also showed significant distortion in both molecules with the carboxylic group almost perpendicular to the backbone plane (85° in both cases). The axial disposition and similar orientation of the hydroxyl and carboxylic functional groups in both terpenoid acids (figure 1*b*) will have an impact on molecule interaction.

Vibrational normal mode calculations were carried out to obtain the theoretical IR spectrum. As can be observed in figure 2*a*, both spectra showed most of the characteristic vibrations that have been previously reported in the literature [6]. Comparison, for each molecule, of the theoretical and experimental FTIR spectra at room temperature showed a high degree of similarity (figure 2*b*). However, the experimental spectra of both molecules displayed a broad and redshifted $\nu$C=O band (approx. 1700 cm$^{-1}$) as well as a stronger $\nu$OH band (approx. 3500 cm$^{-1}$), probably due to environmental humidity. Fernandes *et al.* related the redshift of the $\nu$C=O band to hydrogen bond interaction [6]. Thus, the splitting of this band could be explained as the effect of different C=O environments. In this sense, deconvolution analysis of the $\nu$C=O band resolved a minimum of four contributions (figure 3) indicating that, for the same functional group, at least four different molecular environments were found. Based on these results, it could be assumed that a raw sample of both molecules presents a structure that is the sum of an undetermined number of conformations, with the C=O functional group located in different molecular environments.

## 3.2. Dimeric aggregation of oleanolic and ursolic acids

### 3.2.1. Single hydrogen-bonded dimers

Homodimer analyses were carried out assuming one or two hydrogen bonds between the monomers. In the case of one hydrogen bond, four possible homodimers were studied: head-head (OLE$_{hh}$ and URS$_{hh}$), tail-tail (OLE$_{tt}$ and URS$_{tt}$) and two possible head-tails depending on the participation of the hydrogen (OLE$_{ht}$ and URS$_{ht}$) or oxygen (OLE$_{ht'}$ and URS$_{ht'}$) of the carboxylic functional group in the bond. The proposed oleanolic acid dimers with their respective optimized structures are schematically represented in figure 4. The corresponding ursolic acid dimers are shown in the electronic supplementary material, figure S1.

In order to analyse the stability of the different proposed aggregations, the RBE was calculated for each structure. Using this theoretical parameter, the energetic gain per structural unit after monomer interaction was evaluated. RBE values for dimer aggregation of oleanolic and ursolic acids are shown in table 1. Results indicate that the regular head-tail arrangement (ht) was the most stable aggregation

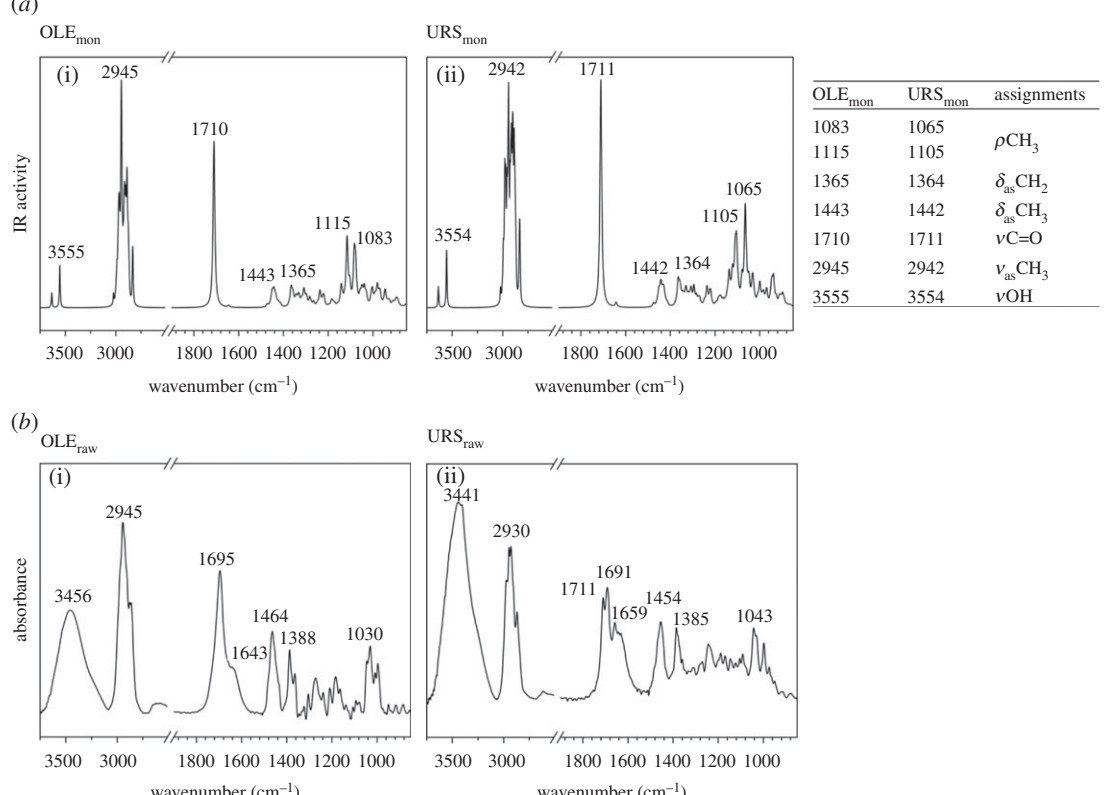

**Figure 2.** (a) Normalized IR spectra calculated for oleanolic (OLE$_{mon}$) and ursolic acid (URS$_{mon}$) together with a table of assignments of selected vibrational IR modes. (b) Experimental FTIR spectra for non-treated samples of oleanolic acid (OLE$_{raw}$) and ursolic (URS$_{raw}$) at room temperature.

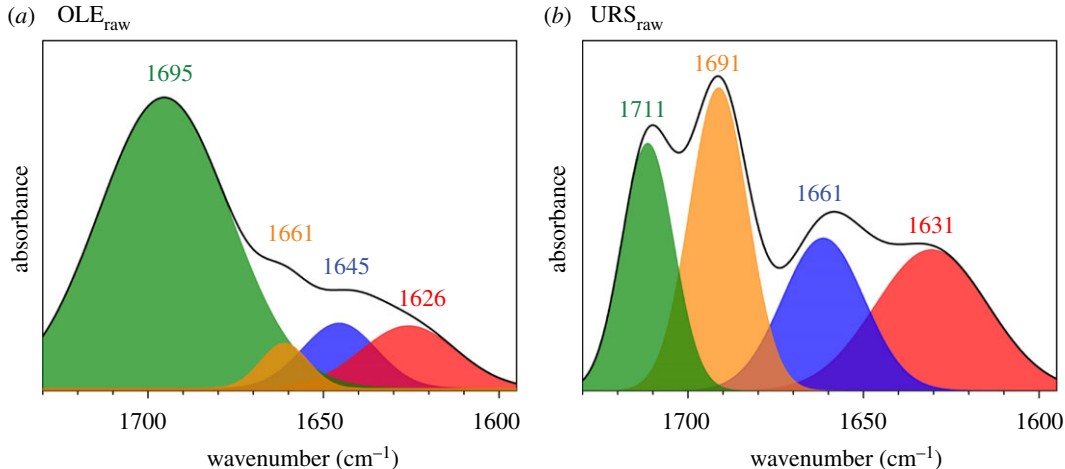

**Figure 3.** Deconvolution of the $v$C=O band spectra of the experimental FTIR (black line) at room temperature for oleanolic (OLE$_{raw}$) (a) and ursolic acids (URS$_{raw}$) (b).

for both isomers. It should be pointed out that, despite ht and ht′ having similar structures, their RBE showed important differences, with ht′ being the least stable of the dimers studied.

According to the literature, the increase in polarity between the oxygen and the hydrogen atoms of a given molecule is directly related with its acidity. Thus, for a higher Mulliken charge difference between the O and H atoms ($\Delta\rho_{OH}$), a stronger hydrogen bond is formed [38]. Therefore, charge distributions around the functional groups of the monomers and homodimers were calculated (electronic supplementary material, figure S2) showing the highest charge difference for the head-tail aggregations (ht). A high charge distribution between the atoms involved in the hydrogen bond ($\Delta\rho_{HB}$) would imply a shorter hydrogen bond and, consequently, a redshift of the $v$C=O band [39–42].

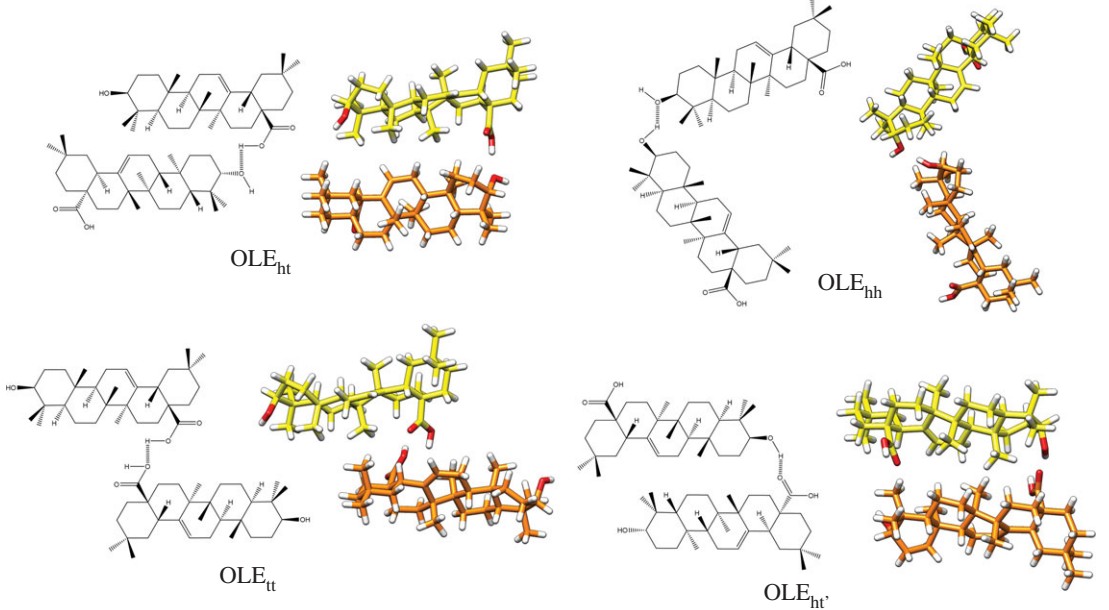

**Figure 4.** Optimized oleanolic acid dimers based on the different location of a single hydrogen bond.

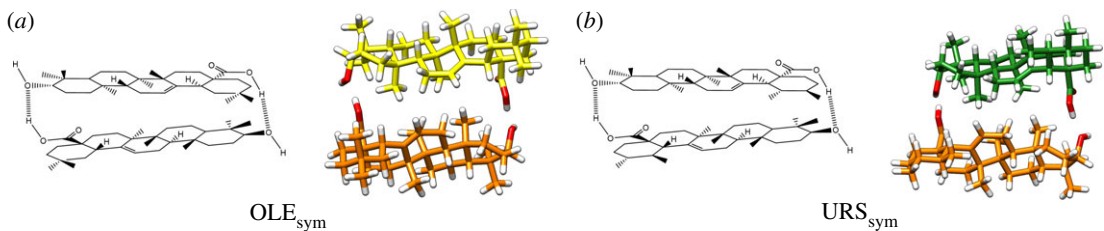

**Figure 5.** Optimized structures of oleanolic (OLE$_{sym}$) (*a*) and ursolic acid (URS$_{sym}$) (*b*) symmetric dimers.

**Table 1.** RBE in kcal mol$^{-1}$ for the dimeric aggregated structures of oleanolic and ursolic acids.

| | dimers | | | | |
|---|---|---|---|---|---|
| RBE (kcal mol$^{-1}$) | ht | hh | tt | ht′ | sym |
| OLE | −10.33 | −6.67 | −8.05 | −4.99 | −14.64 |
| URS | −10.37 | −6.66 | −6.62 | −7.54 | −13.90 |

The effect of the hydrogen bond on the CO functional group was also observed in the theoretical IR spectra for all the dimers studied (electronic supplementary material, figure S3). A splitting of the $\nu$C=O band, due to different environments, can be observed. The eigenvectors extracted for the band located approximately 1700 cm$^{-1}$ (electronic supplementary material, figure S4) confirmed that the redshifted band can be assigned to a CO functional group involved in a hydrogen bond.

### 3.2.2. Doubly hydrogen-bonded dimers

Monomer interaction assuming two hydrogen bonds is only possible in the head-tail orientation as symmetric homodimers (OLE$_{sym}$ and URS$_{sym}$) (figure 5), since the other orientations (hh, tt, ht′) do not support a second hydrogen bond formation.

Considering the energy stabilization that a head-tail hydrogen bond supposes, a lower RBE and higher $\Delta\rho_{HB}$ would be expected in a symmetric dimer as it can be observed in table 1 and electronic supplementary material, figure S2, respectively. The symmetric dimers present C$_2$ symmetry. This implies that the carboxylic group is, from a structural point of view, equivalent in the two monomers.

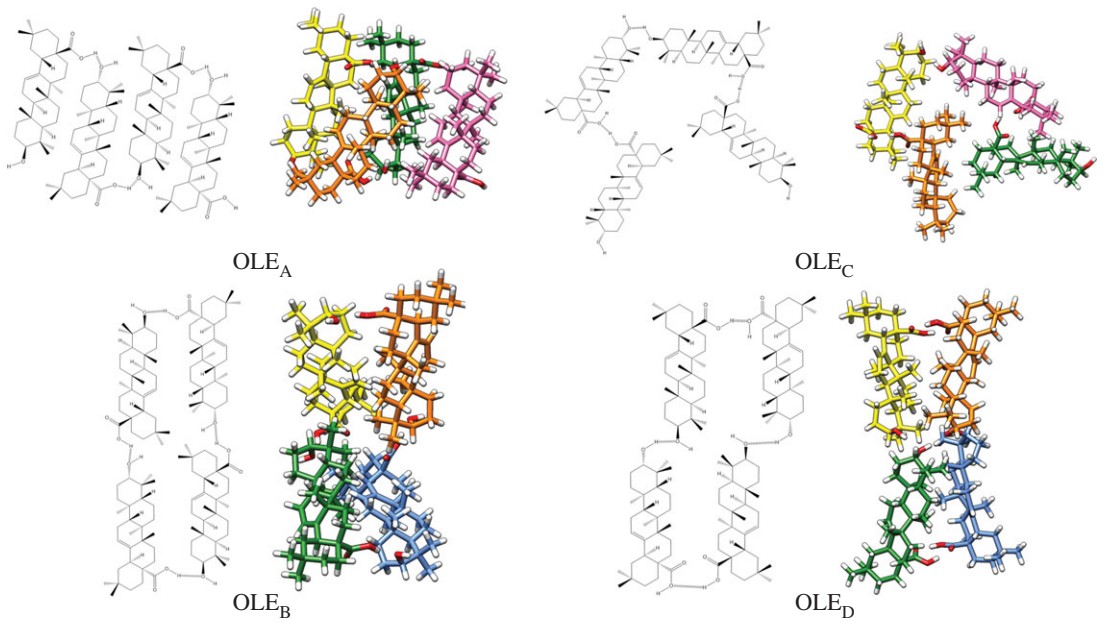

**Figure 6.** Proposed oleanolic acid tetramers (A-D) based on single hydrogen-bonded dimers.

**Table 2.** RBE in kcal mol$^{-1}$ for the tetrameric and higher aggregated structures of oleanolic and ursolic acids.

| RBE (kcal mol$^{-1}$) | A | B | C | D | X | Y | Z | XY | XZ | YZ | XYZ |
|---|---|---|---|---|---|---|---|---|---|---|---|
| OLE | −20.25 | −18.43 | −13.73 | −22.12 | −21.43 | −18.77 | −19.76 | −22.90 | −21.93 | −23.70 | −22.79 |
| URS | −19.54 | −17.64 | −9.13 | −21.63 | −21.30 | −18.60 | −18.16 | −18.87 | −21.90 | −22.41 | −17.48 |

Consequently, the $v$C=O band of the IR spectra did not show splitting (electronic supplementary material, figure S5). Summarizing, these symmetric dimers showed the highest stability, more than those derived from a single hydrogen bond.

## 3.3. Tetrameric aggregation of oleanolic and ursolic acids

### 3.3.1. From single hydrogen-bonded dimers

The single hydrogen-bonded dimers have free functional groups and hence are able to interact with another homodimer. The theoretical analysis of tetramers was carried out assuming different aggregations (head-tail or head-head/tail-tail) and the presence (open) or absence (closed) of free polar functional groups suitable to establish further hydrogen bonds. The proposed oleanolic acid tetramers are shown in figure 6. They are OLE$_A$ (head-tail, open), OLE$_B$ (head-tail, closed), OLE$_C$ (head-head/tail-tail, open) and OLE$_D$ (head-head/tail-tail, closed). The corresponding ursolic acid tetramers (URS$_A$, URS$_B$, URS$_C$ and URS$_D$) are presented in the electronic supplementary material, figure S6.

Analysis of the relative energy stability of the different structures was carried out after the calculation of their corresponding RBE (table 2). An open oligomeric structure is more stable when a head-tail aggregation predominates (OLE$_A$ and URS$_A$), while the closed structures appeared more stable when there is a head-head and tail-tail growth (URS$_D$ and OLE$_D$). Thus, a head-tail growth will more probably form a structure with free functional groups, whereas a head-head and tail-tail growth will tend to form closed tetramers.

Theoretical IR spectra for the proposed tetramers were calculated for both terpenoids (electronic supplementary material, figure S7). As was expected, the closed tetramers, B and D, did not display a carboxylic band at 1711 cm$^{-1}$ but it was instead redshifted, especially in OLE$_D$ and URS$_D$. These results coincide with the absorption at lower frequencies of OLE$_{raw}$ and URS$_{raw}$ (figure 3, red spectrum).

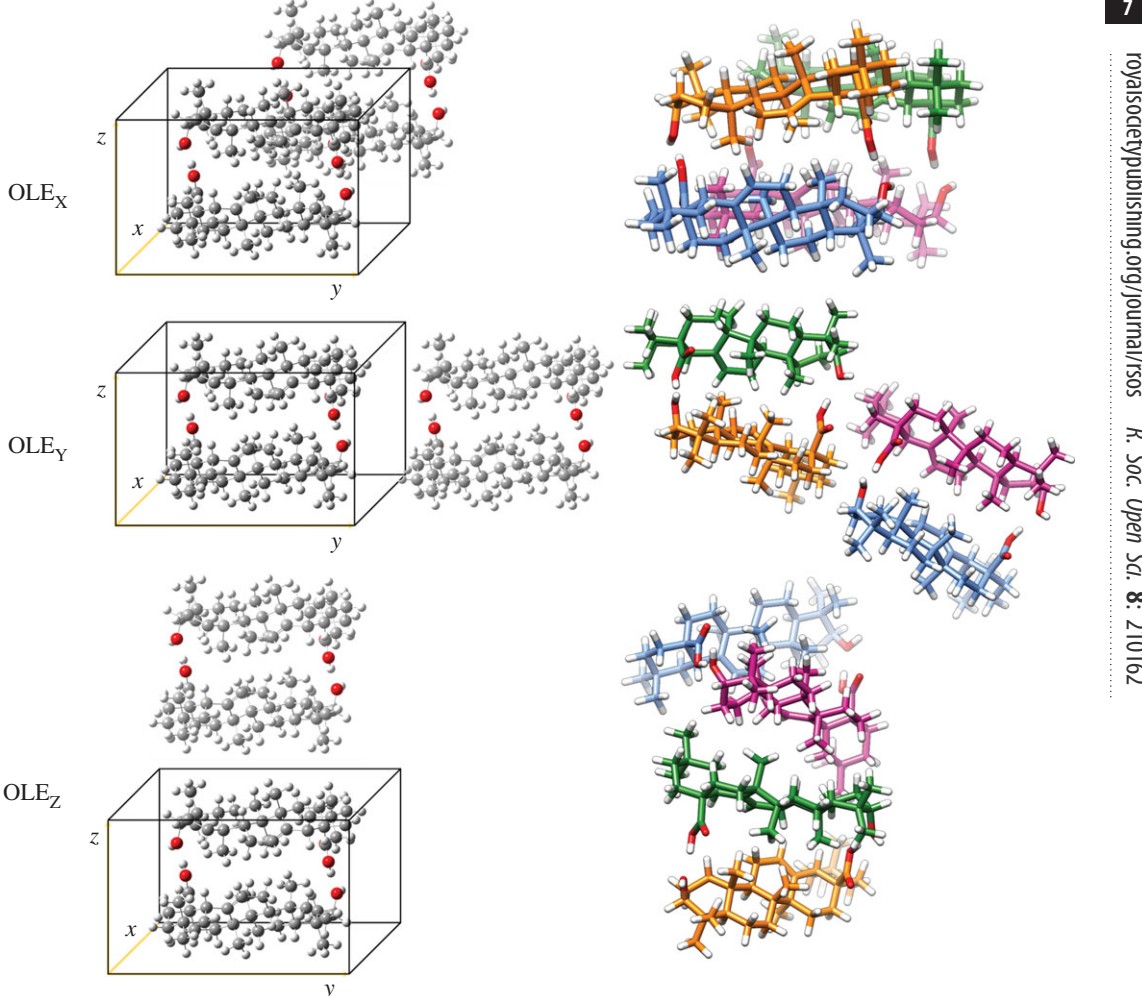

**Figure 7.** Proposed oleanolic acid tretamers (X-Z) based on the symmetric dimer.

### 3.3.2. From doubly hydrogen-bonded dimers

Molecular crystalline growth from symmetric dimers along the X, Y and Z axis ($OLE_X$/$URS_X$, $OLE_Y$/$URS_Y$ and $OLE_Z$/$URS_Z$, respectively) was also considered. The growth scheme and the optimized structures for these aggregates are presented in figure 7 for oleanolic acid, considering $OLE_{sym}$ the unit cell, and in the electronic supplementary material, figure S8 for ursolic acid. Depending on the axis, different growth patterns can be identified and proposed. Thus, $OLE_X$ and $URS_X$ tetramers have a lineal growth; $OLE_Y$ and $URS_Y$ present a bending of the structure while $OLE_Z$ and $URS_Z$ display a helical growth.

As was shown in previous studies, and similarly to the behaviour of amyrin molecules [2], a high tendency of these molecules to assemble by non-electronic interactions was found, meaning a molecule overlapping with no specific attraction force. Consequently, $OLE_X$ presented a more effective stacking based on van der Waals interactions between unit cells compared to $OLE_Y$ and $OLE_Z$ (table 2). Moreover, $OLE_X$ and $URS_X$ have the highest energy stabilization of the symmetric tetramers, very similar to those obtained for the tetramers $OLE_A$/$URS_A$ and $OLE_D$/$URS_D$ derived from single hydrogen-bonded dimers (table 2). Hence, these structures would probably be more abundant in conformational blends of the respective terpenoid acids.

The $OLE_X$/$URS_X$ structure admits a regular growth which is characterized by a non-splitted $\nu$C=O band (electronic supplementary material, figure S9). The hydrogen bond $\nu$C=O stretching band was more redshifted in $OLE_X$/$URS_X$ than in $OLE_{sym}$/$URS_{sym}$, indicating that a more stable assembled structure was found in tetramers [39].

Molecular growth was also analysed after the addition, in different orientations, of further symmetric dimers to the tetrameric structures. Thus, three hexamers ($OLE_{XY}$/$URS_{XY}$, $OLE_{XZ}$/$URS_{XZ}$ and $OLE_{YZ}$/$URS_{YZ}$) and an octamer ($OLE_{XYZ}$/$URS_{XYZ}$) were studied. Their respective RBE showed that the

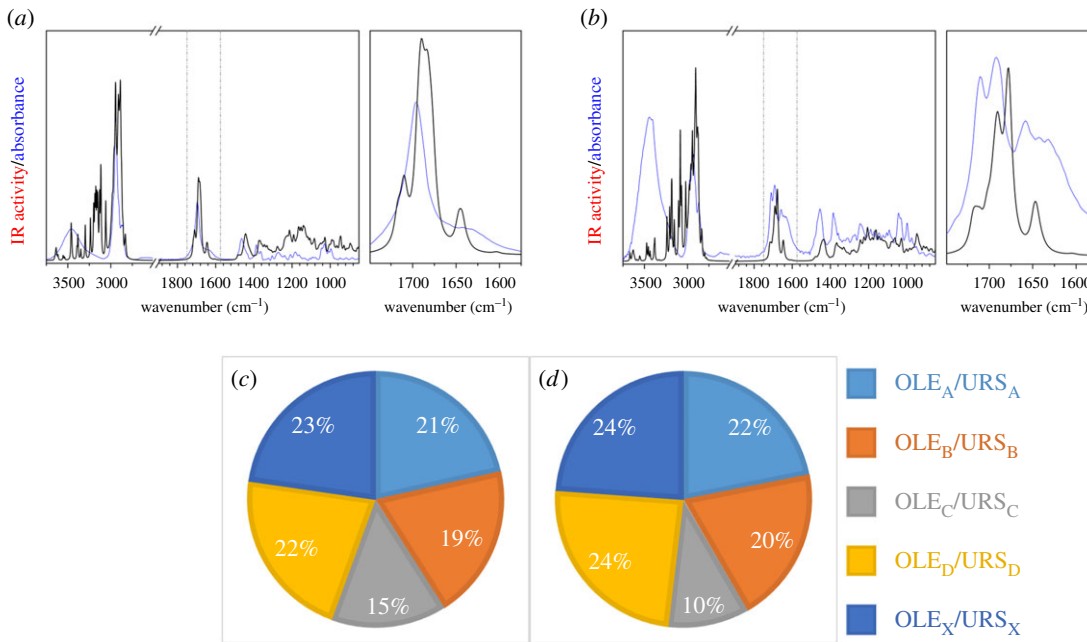

**Figure 8.** Maxwell–Boltzmann weighted theoretical IR spectra of oleanolic (*a*) and ursolic (*b*) tetramers (red) together with the experimental FTIR spectra at room temperature (blue). A zoom of the $\nu$C=O spectral region, delimited by dotted lines, is shown to the right of each IR spectra. (*c,d*) Relative concentration of the different oleanolic and ursolic conformers obtained from their Maxwell–Boltzmann populations.

addition of new dimers, regardless of the orientation, barely improved the energy stability compared to the most stable tetramer $URS_X/OLE_X$, or even displayed a small decrease as in the case of $URS_{XY}$ and the octamer $URS_{XYZ}$ (table 2). These results indicate that a unidirectional crystal growth is expected when these molecules aggregate in a symmetric dimer structure.

Average intermolecular distances between the molecules participating in the different dimer and tetramer arrangements are shown in the electronic supplementary material, table S1. Oleanolic and ursolic symmetric dimers and their corresponding tetramers showed the lowest intermolecular distance compared to the rest of the dimers and tetramers. These calculated intermolecular distances for $OLE_{sym}$ and $OLE_X$ agree well with the experimental X-ray diffraction data reported in the literature for a raw sample of oleanolic acid, where a basal space of 6.5–6.9 Å was determined [6]. Moreover, X-ray diffraction of grape fruit epicuticular waxes [14] that are highly enriched in oleanolic acid showed an average intermolecular distance of 5.7 Å, close to that of $OLE_{sym}$, suggesting that this dimer could be a putative building block in grape waxes.

### 3.3.3. Energy control of the aggregation process

The RBE analysis is based on the difference between the potential energy for each aggregated structure and the potential energy of a single molecule of oleanolic or ursolic acid. However, the entropic contribution to the aggregation reaction, that would favour the process, was not considered. Thus, changes in enthalpy ($\Delta H$), entropy ($\Delta S$) and free energy ($\Delta G$) between the different aggregates of oleanolic and ursolic acids and their corresponding monomers were calculated (electronic supplementary material, table S2). Since $\Delta H$ and $\Delta S$ are negative, a strong entropic control can be considered: a heating of the system involves a less effective crystallization (as $\Delta G$ becomes less negative).

Free energy analysis for the different aggregates showed a similar result to the RBE analysis: monomer aggregation into tetramers, mainly A and D, display higher free energy changes.

### 3.3.4. Molecular structure of raw samples of terpenoid acids

Based on the results obtained, we postulate that a raw sample of each terpenoid would contain a blend of different aggregated structures, most of them studied in the present work. To test this hypothesis, FTIR spectra of both acids were registered at different temperatures (electronic supplementary material, figure S10). As expected, sample heating produced remarkable changes in the $\nu$C=O band, especially a loss of

absorbance at frequencies below 1711 cm$^{-1}$, indicating an important participation of the carboxylic band in the hydrogen bonding interactions responsible for monomer aggregation.

A theoretical spectrum for each terpenoid acid can be obtained based on a Maxwell–Boltzmann population distribution [2,43,44]. This spectrum can be recreated considering the proportional weight of each tetrameric conformer following the expression:

$$\frac{N_i}{N} = \frac{g_i}{e^{(E_i - \mu)/kT}} \rightarrow -\ln\left(\frac{N_i}{N}\right) \propto \frac{E_i}{T},$$

where $N_i$ is the expected number of particles within a given microstate, $N$ is the total number of particles within the system, $g_i$ is the degeneracy of energy level, $E_i$ is the energy that characterizes each of the microstates, $\mu$ is the chemical potential, $k$ is the Boltzmann's constant and $T$ is the temperature of the system.

Figure 8 shows the comparison between the experimental FTIR registered at room temperature and the average theoretical IR spectra obtained after weighting the spectrum of each tetrameric conformer. The Maxwell–Boltzmann weighted spectra presented a better fitting with the experimental FTIR than the spectra of the individual dimers and tetramers previously analysed. This clearly indicates that a blend of different conformational aggregates is the best model to describe raw samples of oleanolic and ursolic acids.

Interestingly, this model provides an explanation for the presence of a relatively high molecular order in the arrangement of terpenoid acids that present at high concentrations in the epicuticular waxes of grape and olive leaves and fruits [14]. To summarize, raw samples of oleanolic and ursolic acids present in the cuticle waxes of plants could be defined as a blend or mixtures of different clusters of different conformers which are aggregated in dimers and tetramers by means of hydrogen bonds and stabilized by non-electrostatic interactions.

## 4. Conclusion

Structural analysis of oleanolic and ursolic acids indicates that they tend to form dimers and tetramers aggregated by hydrogen bonds and stabilized by non-electrostatic interactions. Thus, raw samples of these triterpenoid acids can be described as a blend of clusters of different conformers, most of which have been studied in this work.

These results agree with the previously reported crystalline fraction of triterpenoids present in the epicuticular waxes of several fruits and leaves.

Data accessibility. Additional information concerning this paper is available in the electronic supplementary material and in Dryad Digital Repository: https://doi.org/10.5061/dryad.5mkkwh74x. The data are provided in the electronic supplementary material [45].

Authors' contributions. L.D.M.G.P. and R.C.G.C. carried out the calculations and analyses and wrote the draft manuscript. E.D. and A.H. designed the study and edited the manuscript. All authors gave final approval for publication.

Competing interests. We declare we have no competing interests

Funding. This work was supported by grant no. RTI2018-094277-B/AEI/10.13039/501100011033 from Agencia Estatal de Investigación, Ministerio de Ciencia e Innovación, Spain co-financed by the European Regional Development Fund (ERDF). Open Access funding provided by the Max Planck Society.

Acknowledgements. Luz D.M. Gómez-Pulido is the recipient of a FPI fellowship (BES-2016-078716) from Spanish MINECO co-funded by the European Social Fund. The authors thankfully acknowledge the computing resources, technical expertise and assistance provided by the SCBI (Supercomputing and Bioinformatics) center and Servicios Centrales de Apoyo a la Investigación (SCAI) of the University of Málaga.

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
