## [Peer Review File · Royal Society Open Science]

Review History

RSOS-210162.R0 (Original submission)

Review form: Reviewer 1

Is the manuscript scientifically sound in its present form?

No

Are the interpretations and conclusions justified by the results?

No

Is the language acceptable?

Yes

Do you have any ethical concerns with this paper?

No

Have you any concerns about statistical analyses in this paper?

No

Recommendation?

Major revision is needed (please make suggestions in comments)

Comments to the Author(s)

Heredia and coworkers determine the possible structures of oleanolic acid and ursolic acid using a combination of DFT calculations and IR experiments. The IR spectra calculated from monomeric acid units do not agree well with the experimental spectra. The authors have used different hydrogen-bonded aggregates to match the FTIR spectra.

My comments are as follows:

1. The introduction should be more focused on this work.
2. The authors constructed the theoretical IR spectra using a FWHM of 10 cm⁻¹. Why a uniform FWHM of 10 cm⁻¹ is applicable to all the different conformers/aggregates/ functional groups? The authors need to explain.
3. The authors fitted the experiments FTIR spectra with 4 peaks. The FWHM of the peaks (same color) varies drastically from oleanolic acid and ursolic acid. Given the similarity in the structures of the two molecules, the authors need to explain this huge variation in the FWHM. Are the fits unique? Can we fit the same peak of more/less number of peaks with different FWHM? The authors should show the goodness of the fits by plotting the experimental spectra the fitted spectra in the same plot. The explanation heavily relies on deconvolution.

Review form: Reviewer 2

Is the manuscript scientifically sound in its present form?

No

Are the interpretations and conclusions justified by the results?

No

Is the language acceptable?

No

Do you have any ethical concerns with this paper?

No

Have you any concerns about statistical analyses in this paper?

No

Recommendation?

Major revision is needed (please make suggestions in comments)

Comments to the Author(s)

The manuscript by Heredia and co-workers reports computational investigations in conjunction with IR spectroscopic measurements on Oleanolic and Ursolic acids. The authors have investigated a series of dimeric and tetrameric configurations using DFT. The computed IR spectra of various aggregates are compared with the experimental IR spectra of raw samples to interpret the presence of aggregates in the raw samples. The computations are carried out using the B3LYP functional with an empirical dispersion correction. The reported findings are interesting. However, the manuscript does need some improvement before it could be considered ready for publication. My comments are:

1. The authors interpret that the raw samples consist of dimers and tetramers that are stabilized by way of hydrogen bonds and non-electrostatic interactions. Can the authors describe these non-electrostatic interactions better? The authors may perhaps find it useful to compare the energetics with and without the empirical dispersion corrections. Additionally, energy decomposition analysis may throw some light on the nature of these non-electrostatic interactions.
2. The authors have reported relative binding energies. But, I do not see it defined in the methodology section. It is important to define it for the benefit of the readers.
3. The authors have characterized the hydrogen bond features by way of Mulliken charge analysis. The reported charge differences in Fig. S2 do not contain any units. Can the authors describe the data more comprehensively?
4. From the geometries of the dimers and tetramers, it is clear that the entropic contribution to the energetics might be playing a crucial role. Can the authors assess the relative importance of enthalpic and entropic features in governing the binding of the various aggregates?
5. In the second paragraph of the results section, the authors mention about a red-shifted C-O bond. However, it is not clear that the red-shift is with respect to which position. Can the authors address this better?
6. In the second paragraph of the section on "From doubly hydrogen-bonded dimers", I see that the authors mention about high tendency of the molecules to assemble by non-electronic interactions. What do the authors mean by this?
7. The authors report Maxwell-Boltzmann weighted theoretical spectra towards the end of the manuscript. The equation provided by the authors has Boltzmann constant missing in it! Can the authors carefully revise this?
8. The presentation of the content in the manuscript needs significant improvement. Punctuations are not carefully followed (for example, no space is introduced between numerical value of a parameter and its units) and some sentences need to be rewritten (for example, "...will have an impact in molecule interaction"). In the caption of Fig. 7 and in SI, "tetramers" is misspelt.

Decision letter (RSOS-210162.R0)

Dear Dr González-Cano:

Title: Structure Determination of Oleanolic and Ursolic acids: a Combined DFT/Vibrational Spectroscopy Methodology
Manuscript ID: RSOS-210162

The editor assigned to your manuscript has now received comments from reviewers. We would like you to revise your paper in accordance with the referee and Subject Editor suggestions which can be found below (not including confidential reports to the Editor). Please note this decision does not guarantee eventual acceptance.

Please submit your revised paper before 21-Apr-2021. Please note that the revision deadline will expire at 00.00am on this date. If we do not hear from you within this time then it will be

assumed that the paper has been withdrawn. In exceptional circumstances, extensions may be possible if agreed with the Editorial Office in advance. We do not allow multiple rounds of revision so we urge you to make every effort to fully address all of the comments at this stage. If deemed necessary by the Editors, your manuscript will be sent back to one or more of the original reviewers for assessment. If the original reviewers are not available we may invite new reviewers.

On behalf of the Subject Editor Professor Anthony Stace and the Associate Editor Dr Debashree Ghosh.

RSC Associate Editor:

Comments to the Author:

The reviewers unanimously opine that a major revision is required. And therefore, I suggest that the authors submit a point-wise reply to the reviewers with their revision.

RSC Associate Editor:

Comments to the Author:

(There are no comments.)

Reviewers' Comments to Author:

Reviewer: 1

Comments to the Author(s)

Heredia and coworkers determine the possible structures of oleanolic acid and ursolic acid using a combination of DFT calculations and IR experiments. The IR spectra calculated from

monomeric acid units do not agree well with the experimental spectra. The authors have used different hydrogen-bonded aggregates to match the FTIR spectra.

My comments are as follows:

1. The introduction should be more focused on this work.
2. The authors constructed the theoretical IR spectra using a FWHM of 10 cm⁻¹. Why a uniform FWHM of 10 cm⁻¹ is applicable to all the different conformers/aggregates/ functional groups? The authors need to explain.
3. The authors fitted the experiments FTIR spectra with 4 peaks. The FWHM of the peaks (same color) varies drastically from oleanolic acid and ursolic acid. Given the similarity in the structures of the two molecules, the authors need to explain this huge variation in the FWHM. Are the fits unique? Can we fit the same peak of more/less number of peaks with different FWHM? The authors should show the goodness of the fits by plotting the experimental spectra the fitted spectra in the same plot. The explanation heavily relies on deconvolution.

Reviewer: 2

Comments to the Author(s)

The manuscript by Heredia and co-workers reports computational investigations in conjunction with IR spectroscopic measurements on Oleanolic and Ursolic acids. The authors have investigated a series of dimeric and tetrameric configurations using DFT. The computed IR spectra of various aggregates are compared with the experimental IR spectra of raw samples to interpret the presence of aggregates in the raw samples. The computations are carried out using the B3LYP functional with an empirical dispersion correction. The reported findings are interesting. However, the manuscript does need some improvement before it could be considered ready for publication. My comments are:

1. The authors interpret that the raw samples consist of dimers and tetramers that are stabilized by way of hydrogen bonds and non-electrostatic interactions. Can the authors describe these non-electrostatic interactions better? The authors may perhaps find it useful to compare the energetics with and without the empirical dispersion corrections. Additionally, energy decomposition analysis may throw some light on the nature of these non-electrostatic interactions.
2. The authors have reported relative binding energies. But, I do not see it defined in the methodology section. It is important to define it for the benefit of the readers.
3. The authors have characterized the hydrogen bond features by way of Mulliken charge analysis. The reported charge differences in Fig. S2 do not contain any units. Can the authors describe the data more comprehensively?
4. From the geometries of the dimers and tetramers, it is clear that the entropic contribution to the energetics might be playing a crucial role. Can the authors assess the relative importance of enthalpic and entropic features in governing the binding of the various aggregates?
5. In the second paragraph of the results section, the authors mention about a red-shifted C-O bond. However, it is not clear that the red-shift is with respect to which position. Can the authors address this better?
6. In the second paragraph of the section on "From doubly hydrogen-bonded dimers", I see that the authors mention about high tendency of the molecules to assemble by non-electronic interactions. What do the authors mean by this?
7. The authors report Maxwell-Boltzmann weighted theoretical spectra towards the end of the manuscript. The equation provided by the authors has Boltzmann constant missing in it! Can the authors carefully revise this?
8. The presentation of the content in the manuscript needs significant improvement. Punctuations are not carefully followed (for example, no space is introduced between numerical value of a parameter and its units) and some sentences need to be rewritten (for example, "...will have an impact in molecule interaction"). In the caption of Fig. 7 and in SI, "tetramers" is misspelt.

Author's Response to Decision Letter for (RSOS-210162.R0)

See Appendix A.

Decision letter (RSOS-210162.R1)

Dear Dr González-Cano:

Title: Structure Determination of Oleanolic and Ursolic acids: a Combined DFT/Vibrational Spectroscopy Methodology
Manuscript ID: RSOS-210162.R1

It is a pleasure to accept your manuscript in its current form for publication in Royal Society Open Science. The chemistry content of Royal Society Open Science is published in collaboration with the Royal Society of Chemistry.

On behalf of the Subject Editor Professor Anthony Stace and the Associate Editor Dr Debashree Ghosh.

RSC Associate Editor
Comments to the Author:
(There are no comments.)

Reviewer(s)' Comments to Author:

Appendix A

Reviewer: 1

Comments to the Author(s)

Heredia and coworkers determine the possible structures of oleanolic acid and ursolic acid using a combination of DFT calculations and IR experiments. The IR spectra calculated from monomeric acid units do not agree well with the experimental spectra. The authors have used different hydrogen-bonded aggregates to match the FTIR spectra.

My comments are as follows:

1. The introduction should be more focused on this work.

The ms focuses in the arrangement of oleanolic acid and ursolic acid within a raw sample in order to explain their behavior within the plant cuticle and also their physicochemical properties. In this sense, the introduction is focused on a description of the structure and composition of the plant cuticle, the location of these acids within the cuticle, as well as some aspects of the structure and chemistry of triterpenoid acids and their uses in different fields. Nevertheless, we have simplified the text in the introduction.

2. The authors constructed the theoretical IR spectra using a FWHM of 10 cm⁻¹. Why a uniform FWHM of 10 cm⁻¹ is applicable to all the different conformers/aggregates/functional groups? The authors need to explain.

Theoretical spectra were built with a standard FWHM of 10 cm⁻¹. This parameter didn't affect the conclusions of the study, based on the wavenumber and relative intensity of the IR band spectra.

3. The authors fitted the experiments FTIR spectra with 4 peaks. The FWHM of the peaks (same color) varies drastically from oleanolic acid and ursolic acid. Given the similarity in the structures of the two molecules, the authors need to explain this huge variation in the FWHM. Are the fits unique? Can we fit the same peak of more/less number of peaks with different FWHM? The authors should show the goodness of the fits by plotting the experimental spectra the fitted spectra in the same plot. The explanation heavily relies on deconvolution.

The aim of the study is to investigate the ability of these molecules to form different aggregates based on the relative position of the monomers looking for the formation of hydrogen bonds. The first step is to point out that in the experimental FTIR spectra the ν C=O band is splitted in different contributions. The aim of the deconvolution represented in this work is to demonstrate the existence and the shifting of these contributions. The reviewer is right mentioning this point. In this sense the ms has been updated in order to clarify it. The Figure 3 has been modified including the corresponding experimental spectra as a black line.

Reviewer: 2

Comments to the Author(s)

The manuscript by Heredia and co-workers reports computational investigations in conjunction with IR spectroscopic measurements on Oleanolic and Ursolic acids. The authors have investigated a series of dimeric and tetrameric configurations using DFT. The computed IR spectra of various aggregates are compared with the experimental IR spectra of raw samples to interpret the presence of aggregates in the raw samples. The computations are carried out using the B3LYP functional with an empirical dispersion correction. The reported findings are interesting. However, the manuscript does need some improvement before it could be considered ready for publication. My comments are:

1. The authors interpret that the raw samples consist of dimers and tetramers that are stabilized by way of hydrogen bonds and non-electrostatic interactions. Can the authors describe these non-electrostatic interactions better? The authors may perhaps find it useful to compare the energetics with and without the empirical dispersion corrections. Additionally, energy decomposition analysis may throw some light on the nature of these non-electrostatic interactions.

Non-electrostatic interactions are forces not related with electronic and polarizable effects (i.e. hydrogen bonds). Calculation and comparison of the OLE_{ht} and URS_{ht} dimers with (RBE) and without empirical dispersion correction (RBE*) clarify the importance of these non-electrostatic interactions (E_{int}):

	RBE (kcal·mol ⁻¹)	RBE* (kcal·mol ⁻¹)	E _{int} (kcal·mol ⁻¹)
OLE _{ht}	-10.33	-4.44	-5.89
URS _{ht}	-10.37	-4.41	-5.96

2. The authors have reported relative binding energies. But, I do not see it defined in the methodology section. It is important to define it for the benefit of the readers.

The methodology section has been modified providing more information on the Relative Binding Energies (RBE) calculation.

3. The authors have characterized the hydrogen bond features by way of Mulliken charge analysis. The reported charge differences in Fig. S2 do not contain any units. Can the authors describe the data more comprehensively?

The Mulliken atomic charge is an electronic population analysis that considers the electron density belonging in an atomic orbital. Hence, it is dimensionless.

4. From the geometries of the dimers and tetramers, it is clear that the entropic contribution to the energetics might be playing a crucial role. Can the authors assess the relative importance of enthalpic and entropic features in governing the binding of the various aggregates? We have found this aspect really interesting and we have updated the ms considering not only the stability energy (RBE) but also the enthalpy (ΔH), entropy (ΔS) and free energy (ΔG) changes obtained after oleanolic and ursolic acid aggregation from their monomers. The conclusions derived from these analyses remain identical to those previously obtained.

5. In the second paragraph of the results section, the authors mention about a red-shifted C-O bond. However, it is not clear that the red-shift is with respect to which position. Can the authors address this better?

The redshifting mentioned in the manuscript is related to the $\nu_{\text{C=O}}$ band of a carboxylic group free of hydrogen bonds (at 1711cm^{-1}). The ms has been updated to clarify this point.

6. In the second paragraph of the section on "From doubly hydrogen-bonded dimers", see that the authors mention about high tendency of the molecules to assemble by non-electronic interactions. What do the authors mean by this?

The non-electronic interactions refer to the molecules overlapping interaction with no specific attraction force. Following the reviewer's comment, the text has been revised.

7. The authors report Maxwell-Boltzmann weighted theoretical spectra towards the end of the manuscript. The equation provided by the authors has Boltzmann constant missing in it! Can the authors carefully revise this?

In the ms, the Maxwell-Boltzmann population distribution is taken into consideration in order to obtain a relationship between the weight of a concrete microstate (a conformer) in the overall energy of the system. The expression in the text is not a formal equation but an indication of the direct proportion or relationship between some parameters and certain physicochemical constants. However, following the reviewer recommendation we have corrected the text to include a proper distribution function.

8. The presentation of the content in the manuscript needs significant improvement. Punctuations are not carefully followed (for example, no space is introduced between numerical value of a parameter and its units) and some sentences need to be rewritten (for example, "...will have an impact in molecule interaction"). In the caption of Fig. 7 and in SI, "tetramers" is misspelt.

The ms and the ESI have been modified in order to improve the punctuation and spelling.